**Data Availability Statement:** All relevant data are within the paper and its Supporting Information files.

**Funding:** The author(s) received no specific funding for this work.

# Patient perspectives of antiretroviral pharmacy services: A cross-sectional cohort study

Yadi Liu[1], Elizabeth Lyden[2], Renae Furl[3], Joshua P. Havens[1,3]*

1 University of Nebraska Medical Center, College of Pharmacy, Omaha, NE, United States of America,
2 University of Nebraska Medical Center, College of Public Health, Omaha, NE, United States of America,
3 University of Nebraska Medical Center, College of Medicine, Omaha, NE, United States of America

* jhavens@unmc.edu

## Abstract

### Background

Adherence to antiretroviral therapy (ART) remains the main predictor of sustained HIV virologic suppression for people with HIV (PWH). Mail-order pharmacy services are often offered to patients as an alternative option to traditional pharmacy services. Some payers mandate ART to be dispensed from specific mail-order pharmacies regardless of patient choice complicating ART adherence for patients affected by social disparities. Yet, little is known about patient perspectives regarding mail-order mandates.

### Methods

Eligible patients of the HIV program at University of Nebraska Medical Center with experience receiving ART from both a local and mail-order pharmacy were invited to complete a 20-question survey with three core sections: experiences/perspectives on local and mail-order pharmacy settings; pharmacy attributes rankings; and pharmacy preference. Paired t-tests and Mann-Whitney tests were used to compare the agreement scores of pharmacy attributes.

### Results

Sixty patients (N = 146; 41.1%) responded to the survey. Mean age was 52 years. Most were male (93%) and White (83%). The majority of participants were on ART for HIV treatment (90%) and 60% were using mail-order pharmacies for their prescription services. Significant scoring differences (p<0.05) were observed for all pharmacy attributes favoring local pharmacies. Refilling ease was the most important attribute noted. More respondents (68%) preferred local pharmacies versus mail-order pharmacies. Payer associated mail-order pharmacy mandates were experienced by 78% with half believing the mandates impacted their medical care negatively.

### Conclusions

In this cohort study, respondents preferred local pharmacies compared to mail-order pharmacy for ART prescription services and noted ease of refilling as the most important

**Competing interests:** The authors have declared that no competing interests exist.

pharmacy attribute. Two-thirds of respondents believed mail-order pharmacy mandates negatively affected their health. Insurance payers should consider the removal of mail-order pharmacy mandates to allow patient choice of pharmacy, which may help remove barriers to ART adherence and improve long-term health outcomes.

## Introduction

Adherence to antiretroviral therapy (ART) among people with HIV (PWH) is essential to sustain viral suppression, reduce the development of HIV drug resistance, and improve long-term health outcomes [1]. Similarly, people taking ART for HIV pre-exposure prophylaxis (PrEP) require high levels of adherence to prevent HIV infection [2,3]. Despite ART/PrEP options offering better tolerability and lower pill burden, some patients still struggle with adherence [4]. Reasons for ART non-adherence among PWH and those on PrEP are vast and include forgetfulness, substance use, medication access/cost, stigma, and other social factors such as housing instability, poverty, etc. [5–10].

Various strategies have shown to improve ART adherence, such as smartphone applications and blister packages [11,12]. An available adherence option to patients is mail-order pharmacy services which may be more convenient, alleviate stigma concerns, and remove transportation to the pharmacy as a barrier to treatment adherence [13]. Mailing medication directly to patients has been associated with increased adherence and improved clinical outcomes in other disease states, such as cancer and diabetes [14,15].

Due to high cost and complexity, ART regimens are increasingly designated by payers as specialty medications on insurance formularies. Specialty pharmacies often facilitate dispensations of complex, high cost, high touch specialty medications and provide adherence support (e.g. counseling services, medication access assistance, refill reminders, etc.). While higher ART adherence rates among PWH have been observed at specialty pharmacies [16,17], recent studies found no benefits in viral suppression rates among PWH utilizing mail-order pharmacy services [9,18]. Some insurance payers mandate members to use specific specialty mail-order pharmacies for their specialty medications that often includes ART [19]. Typically, such mandates allow for cost savings for the payer but disallow patient pharmacy choice potentially complicating medication adherence for PWH affected by social disparities such as housing instability [5,20], inconsistent access to a telephone or other electronic communication devices, social isolation [21], and language barriers.

Little is known about patient perspectives regarding payer administered mail-order mandates and pharmacy services for ART among PWH or those on PrEP. Further insight into patients' perspectives on pharmacy settings may help remove barriers to ART adherence and improve long-term health outcomes. In this study, we aimed to evaluate patient perspectives on various aspects of prescription services among participants with experience receiving pharmacy services from both a local pharmacy and a mail-order pharmacy.

## Methods

### Study design, data collection, and inclusion criteria

We conducted a prospective, cross-sectional survey study of patients receiving care at the University of Nebraska Medical Center HIV clinic, the Specialty Care Center (SCC). Inclusion criteria for study entry were: 1) age greater than 19 years old; 2) retained in care for ≥6 months;

3) prescribed ART for HIV treatment or PrEP; and 4) experience receiving pharmacy services from both a local pharmacy and mail-order pharmacy. Participants were excluded if they did not complete the survey entirely, were participating in an industry study, or non-English speaking.

Local pharmacies were defined as an "in-town, local pharmacy, where patients had the option to pick up medications" whereas a mail-order pharmacy was defined as an "out-of-town pharmacy, where the medications had to be mailed to patients' home".

We designed a survey to measure the patients' perspectives and preferences of pharmacy services. The survey consisted of 20 questions divided into three sections: experiences/perspectives on both pharmacy settings, ranking of pharmacy attributes, and pharmacy preference. The first section used a 5-point Likert scale (1 [strongly disagree] to 5 [strongly agree]) to inquire about patient satisfaction, general pharmacy experience, medication ordering/access/dispensing, and clinical services for both pharmacy settings. In this section, the respondents were asked to score each attribute separately for both pharmacy settings (local and mail-order). The second section asked participants to rank the following pharmacy service attributes in order of importance: trust in and relationship with the pharmacy staff, ease of ordering medication refills, additional adherence services, expedited medication delivery (couriered or overnight), and access to the pharmacist when needed. The last section assessed overall pharmacy preference, whether participants had ever been mandated to use a specific mail-order pharmacy, and whether they believed the mandate negatively affected their medical care.

All eligible participants were contacted in-person during a scheduled clinic visit or through electronic communication through the electronic health record, Epic (Verona, Wisconsin). Surveys were disseminated via two methods: REDCap [22,23] or via paper. Viral suppression status and ART refill histories were collected for each respondent. Adherence was measured using the percentage of days covered (PDC) metric calculated as the total number of tablets dispensed over the past year divided by 365 days multiplied by 100%.

## Patient consent statement

The University of Nebraska Medical Center's Institutional Review Board (IRB #741-21-EP) approved this study. Participants provided written informed consent prior to survey initiation.

## Statistical analysis

Descriptive statistics were used to summarize patient demographics and baseline characteristics. Counts and percentages were used for categorical data with median (range) or mean (standard deviation [SD]) used for continuous data. The Mann-Whitney test was used to compare the median value of agreement questions based on preference (local vs. mail-order) and requirement (no mail-order pharmacy mandate vs. mail-order pharmacy mandate). Fisher's exact test was used to evaluate the association demographic characteristics and preference. All analyses were done using SAS Version 9.4, SAS Institute Inc., Cary, NC. A p-value of less than 0.05 was considered statistically significant.

## Results

Sixty patients (N = 146; 41.1%) responded to the survey. The median age was 52 (range, 28–77) years. Most of participants were male (93%), White (83%), not Hispanic (95%), and commercially insured (83%). Six respondents (10%) were on ART for PrEP. Mean adherence, by percentage of days covered, to ART was high overall at 91%. Among PWH, 50 (93%) and 54 (100%) respondents had HIV RNA <50 and <200 copies/mL, respectively. At the time of

**Table 1. Baseline characteristics.**

| n (%) | Pharmacy At Time of Survey | | Overall (N = 60) |
|---|---|---|---|
| | Local (n = 25) | Mail-Order (n = 35) | |
| Age, median (range) | 51.0 (30–71) | 52.0 (28–77) | 52.0 (28–77) |
| Sex | | | |
| • Male | 23 (92.0) | 32 (91.4) | 56 (93.3) |
| • Female | 2 (8.0) | 3 (8.6) | 4 (6.7) |
| Race | | | |
| • White | 20 (80.0) | 31 (88.6) | 50 (83.3) |
| • Black | 5 (20.0) | 3 (8.6) | 9 (15.0) |
| • Asian | 0 (0.0) | 1 (2.8) | 1 (1.7) |
| Ethnicity | | | |
| • Hispanic | 1 (4.0) | 2 (5.7) | 3 (5.0) |
| • Not Hispanic | 24 (96.0) | 33 (94.3) | 57 (95.0) |
| Insurance Coverage | | | |
| • Commercial | 20 (80.0) | 28 (80.0) | 48 (80.0) |
| • Medicare | 2 (8.0) | 5 (14.4) | 7 (11.6) |
| • Medicaid | 3 (12.0) | 1 (2.8) | 4 (6.7) |
| • VA/Government | 0 (0.0) | 1 (2.8) | 1 (1.7) |
| Treatment | | | |
| • HIV Treatment | 25 (100.0) | 29 (82.9) | 54 (90.0) |
| • HIV Prevention (PrEP) | 0 (0.0) | 6 (17.1) | 6 (10.0) |
| Adherence (PDC, 1 year), mean (SD) | 93.0 (0.2) | 90.0 (0.2) | 91.3 (0.2) |
| Last HIV RNA <50 copies/mL[a] | 23 (92.0) | 27 (93.1) | 50 (92.6) |
| Last HIV RNA <200 copies/mL[a] | 25 (100.0) | 29 (100.0) | 54 (100.0) |

[a]Viral suppression based on a total N of 54 (i.e. total number of respondents with HIV).

Definitions: PWH–people with HIV; PDC–percentage of days covered; PrEP–HIV pre-exposure prophylaxis; VA–Veterans Administration.

survey completion, 58% were receiving prescription service from a mail-order pharmacy. **Table 1** summarizes baseline characteristics.

## Perspectives on pharmacy services

Respondents were asked 16 questions regarding their perspectives on pharmacy services for both local and mail-order pharmacy settings. Specific focuses included general pharmacy experience, pharmacy settings, and medication ordering/access/dispensing and clinical services. Agreement scores were consistently significantly higher for the local pharmacy setting (all p<0.05). A descriptive summary of mean pharmacy attribute scoring for both local and mail-order pharmacy settings is described in **Fig 1**.

## Pharmacy attribute rankings

Respondents ranked five components: trust in pharmacy, ease of refilling, additional services provided, delivery methods and pharmacist accessibility. Respondents ranked "ease of ordering medication refills" as the most important attribute; whereas, "additional services to help me take my medications as prescribed" was ranked least important (**Fig 2**).

## Mail-order pharmacy mandates and respondent preferences

Forty-seven participants (78%) reported their insurance had mandated use of a specific pharmacy for their ART. Further, 43% (20/47) of these respondents believed being mandated to use a specific mail-order pharmacy impacted their medical care. More respondents preferred a

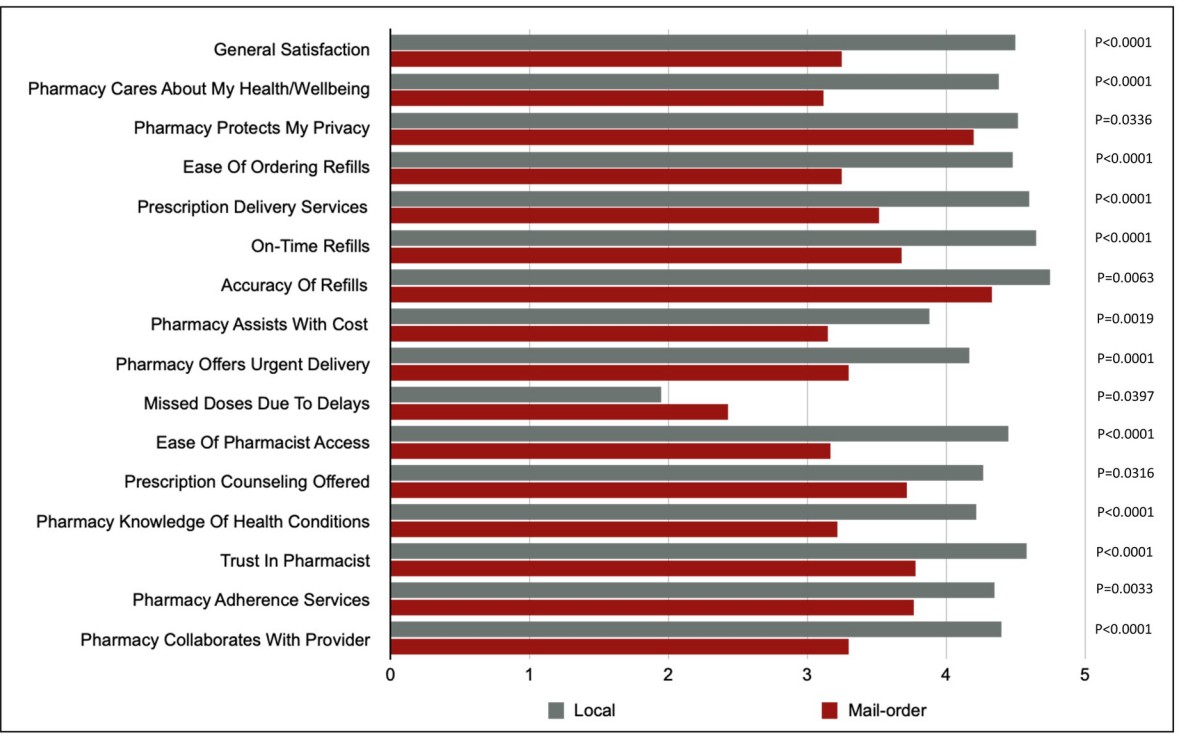

**Fig 1. Comparisions of respondent pharmacy attribute scoring.** Agreement scale: 1 (strongly disagree) to 5 (strongly agree).

local pharmacy (n = 41, 68%) (**Fig 3**). However, no significant differences were observed when comparing respondent pharmacy preference based on whether they had experienced a mail-order pharmacy mandate (p = 0.2604) or by baseline characteristics (**S1 Table**). No significant differences in pharmacy preference was observed based on respondent demographics.

Mean scores for pharmacy attributes for both pharmacy settings were individually compared by respondent pharmacy preference and whether they had experienced a mail-order pharmacy mandate (**Table 2**). When comparing attribute scores for the local pharmacy setting, significant higher differences were observed for respondents preferring a local pharmacy compared to mail-order pharmacy for refill ordering ease (p = 0.0098), pharmacy delivery services (p = 0.0292), on-time refills (p = 0.0004), prescription counseling offered (p = 0.0134), and pharmacy knowledge of health condition (p = 0.0107). Only attribute scoring for pharmacy delivery services (p = 0.0233) was significantly higher when comparing local pharmacy attribute scores among respondents who had experienced a mail-order pharmacy mandate vs not.

Alternatively, when comparing pharmacy attribute scores for the mail-order pharmacy setting, significant lower differences were observed for general satisfaction (p<0.0001), pharmacy cares about my health/wellbeing (p = 0.0010), privacy (p = 0.0441), refill ordering ease (p = 0.0002), pharmacy delivery services (p = 0.0251), on-time refills (p = 0.0208), offering urgent prescription delivery (p = 0.0004), ease of pharmacist access (p = 0.0238), trust in pharmacist (p = 0. 0019) and pharmacy collaboration with providers (p = 0.0067) for respondents preferring a local pharmacy vs mail-order pharmacy. Similarly, significantly lower differences were observed in pharmacy attribute score comparisons for the mail-order setting for general satisfaction (p = 0.0090), pharmacy cares about my health/wellbeing (p = 0.0037), refill

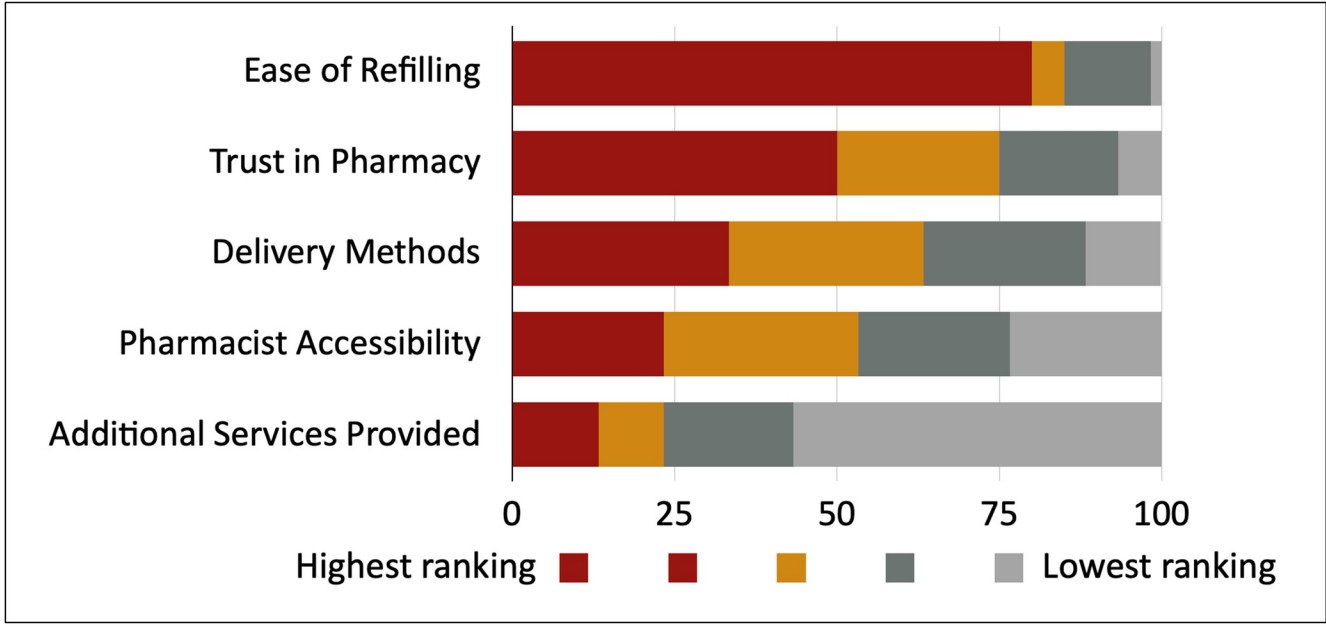

**Fig 2. Respondent rankings of pharmacy attributes.**

ordering ease (p = 0.0055), offering urgent prescription delivery (p = 0.0167), ease of pharmacist access (p = 0.0013), pharmacy knowledge of health condition (p = 0.0169), and pharmacy collaboration with providers (p = 0.0290) among respondents who had experienced a mail-order pharmacy mandate vs not.

## Discussion

In this cross-sectional, cohort study, we surveyed PWH and patients on PrEP regarding their perspectives/experience receiving pharmacy services from both local and mail-order pharmacy settings. Overall, significantly, higher scores were observed for the local pharmacy setting and the majority of respondents preferred the local pharmacy setting. Respondents believed ease of ordering ART refills was the most important pharmacy attribute regardless of pharmacy setting.

While mail-order pharmacies are thought to improve access to medications and ultimately increase adherence [14–17], many PWH exhibit higher rates of social disparities (e.g. unstable housing, poverty, unemployment, access to healthcare, etc.) [24–26], which may further complicate successful ART dispensations from a mail-order pharmacy for this population. Alternatively, local pharmacies have been shown to provide personal touch [27]. Pharmacists in the local pharmacy setting are often seen by patients as an approachable healthcare resource within the community and face-to-face counseling between pharmacists and patients allows for personalized consultation, strong relationship building, and has been shown to improve medication adherence [28]. Further, patients are often provided more flexibility in

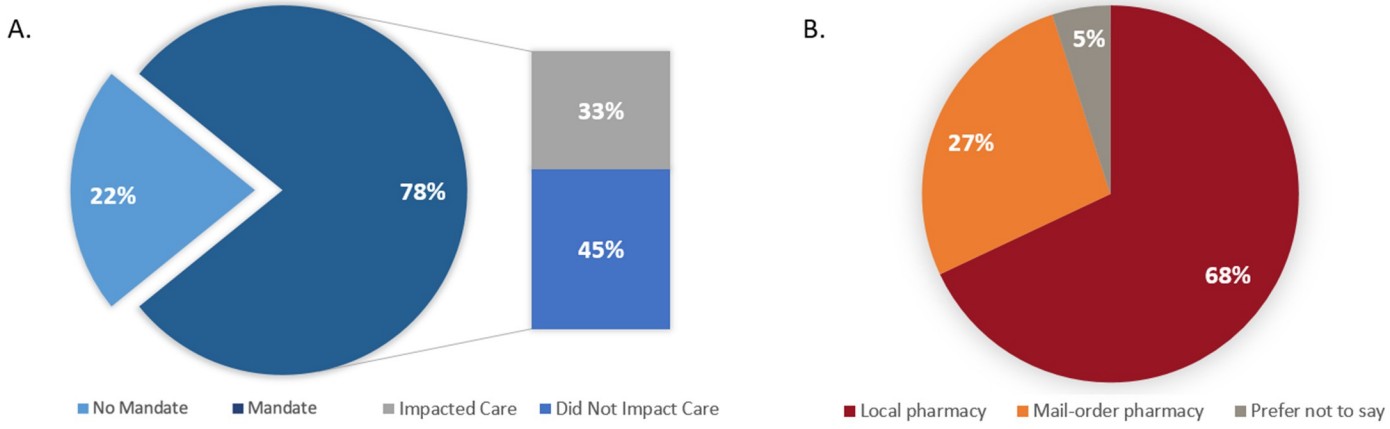

**Fig 3. Payer mail-order pharmacy mandate experience and pharmacy preference.** *A*, Respondent mail-order pharmacy mandate experience and beliefs on impact of care. *B*, Respondent pharmacy setting preference.

prescription access at the local pharmacy whereby they may choose to either pick-up their prescription directly or have it delivered by either mail or couriered service. This theoretically removes some social disparity-related adherence barriers such as transportation and unstable housing. Importantly, our data suggests respondents believed they missed ART doses due to prescription refill delays more frequently with mail-order pharmacy based on attribute scoring (**Fig 1**). This is consistent with a Veterans Affairs cohort study by Desai and colleagues, which found 47% of participants reporting prescription delivery delays leading to missed ART dosing while using mail-order service [29].

Most respondents (68%) reported a preference for the local pharmacy for prescription services. Respondents scoring generally followed their pharmacy setting preference. However, no significant differences in pharmacy preference was observed based on demographic characteristics or mail-order pharmacy mandate experience (**S1 Table**). Importantly, respondents ranked "ease of refills" as the most important pharmacy attribute and this was further evident in respondent mean scoring comparisons for both local and mail-order pharmacy settings (**Table 2**) based on preference (local pharmacy scoring: local preference, 4.66 vs mail-order preference, 3.94 [p = 0.0098]; mail-order scoring: local preference, 2.71 vs mail-order preference, 4.38 [p = 0.0002]). Similarly, "prescription delivery services" and "on-time refills" were notable for differences in scoring comparisons. Thus, ensuring a streamlined, easy ordering/refilling process and/or removing any associated barriers should be prioritized by pharmacies regardless of setting. Notably, respondents ranked "additional adherence services" least important and this was similarly evident in scoring comparisons with no significant differences

**Table 2. Local vs. mail-order pharmacy attribute scoring comparisons by pharmacy preference and mandate experience.**

| Pharmacy Attribute | Local Pharmacy Rating Scores (means) | | | | | |
| --- | --- | --- | --- | --- | --- | --- |
| | Local Preference (n = 41) | Mail-Order Preference (n = 16) | P-Value | No Mandate (n = 13) | Mandate (n = 47) | P-Value |
| General Satisfaction | 4.54 | 4.31 | 0.2293 | 4.38 | 4.53 | 0.7639 |
| Pharmacy Cares About My Health/ Wellbeing | 4.49 | 4.13 | 0.2167 | 4.23 | 4.43 | 0.3302 |
| Pharmacy Protects My Privacy | 4.61 | 4.31 | 0.0519 | 4.23 | 4.60 | 0.1665 |
| Ease Of Ordering Refills | **4.66** | **3.94** | **0.0098** | 4.00 | 4.62 | 0.1245 |
| Prescription Delivery Services | **4.76** | **4.25** | **0.0292** | **4.23** | **4.70** | **0.0233** |
| On-Time Refills | **4.80** | **4.19** | **0.0004** | 4.46 | 4.70 | 0.7644 |
| Accuracy of Refills | 4.78 | 4.63 | 0.2364 | 4.62 | 4.79 | 0.4797 |
| Pharmacy Assists With Cost | 4.15 | 3.25 | 0.0982 | 3.62 | 3.96 | 0.4133 |
| Pharmacy Offers Urgent Delivery | 4.15 | 4.19 | 0.5553 | 4.15 | 4.17 | 0.7110 |
| Missed Doses Due To Delays | 1.93 | 2.19 | 0.2559 | 2.00 | 1.94 | 1.0000 |
| Ease Of Pharmacist Access | 4.54 | 4.13 | 0.0569 | 4.15 | 4.53 | 0.1324 |
| Prescription Counseling Offered | **4.49** | **3.63** | **0.0134** | 3.77 | 4.40 | 0.0639 |
| Pharmacy Knowledge of health condition | **4.46** | **3.63** | **0.0107** | 4.00 | 4.28 | 0.2629 |
| Trust In Pharmacist | 4.61 | 4.50 | 0.1074 | 4.38 | 4.64 | 0.1188 |
| Pharmacy Adherence Services | 4.49 | 3.94 | 0.0792 | 4.23 | 4.38 | 0.7034 |
| Pharmacy Collaborates With Provider | 4.46 | 4.13 | 0.2104 | 4.31 | 4.43 | 0.5128 |
| Pharmacy Attribute | Mail-Order Pharmacy Rating Scores (means) | | | | | |
| | Local Preference (n = 41) | Mail-Order Preference (n = 16) | P-Value | No Mandate (n = 13) | Mandate (n = 47) | P-Value |
| General Satisfaction | **2.68** | **4.44** | **<0.0001** | **4.15** | **3.00** | **0.0090** |
| Pharmacy Cares About My Health/Wellbeing | **2.63** | **4.13** | **0.0010** | **4.23** | **2.81** | **0.0037** |
| Pharmacy Protects My Privacy | **4.00** | **4.69** | **0.0441** | 4.38 | 4.15 | 0.7185 |
| Ease Of Ordering Refills | **2.71** | **4.38** | **0.0002** | **4.31** | **2.96** | **0.0055** |
| Prescription Delivery Services | **3.20** | **4.13** | **0.0251** | 4.23 | 3.32 | 0.0645 |
| On-Time Refills | **3.29** | **4.44** | **0.0208** | 4.23 | 3.53 | 0.1705 |
| Accuracy of Refills | 4.12 | 4.75 | 0.1172 | 4.69 | 4.23 | 0.3627 |
| Pharmacy Assists With Cost | 3.07 | 3.25 | 0.6403 | 3.69 | 3.00 | 0.1588 |
| Pharmacy Offers Urgent Delivery | **2.83** | **4.31** | **0.0004** | **4.15** | **3.06** | **0.0167** |
| Missed Doses Due To Delays | 2.49 | 2.56 | 0.8174 | 2.00 | 2.55 | 0.2504 |
| Ease Of Pharmacist Access | **2.85** | **3.75** | **0.0238** | **4.23** | **2.87** | **0.0013** |
| Prescription Counseling Offered | 3.56 | 4.06 | 0.3128 | 4.00 | 3.64 | 0.5828 |
| Pharmacy Knowledge of health condition | 2.98 | 3.75 | 0.0503 | **4.08** | **2.98** | **0.0169** |
| Trust In Pharmacist | **3.39** | **4.63** | **0.0019** | 4.46 | 3.60 | 0.0523 |
| Pharmacy Adherence Services | 3.56 | 4.13 | 0.1283 | 4.31 | 3.62 | 0.0756 |
| Pharmacy Collaborates With Provider | **2.90** | **4.06** | **0.0067** | **4.15** | **3.06** | **0.0290** |

Respondents were asked to score pharmacy attributes (Scale: 1 = strongly disagree to 5 = strongly agree) for each pharmacy setting (i.e. local and mail-order). A, Comparing *local* pharmacy setting attribute scores by patient pharmacy preference and insurance pharmacy mandate experience; B, Comparing *mail-order* pharmacy setting attribute scores by patient pharmacy preference and insurance pharmacy mandate experience.

observed. This is especially important given payers high prioritization of medication adherence for quality ratings [30,31]. However, our patients displayed high levels of ART adherence by PDC (>90%) and may have not needed additional adherence services to help them take their medication.

The majority of our respondents (78%) reported experiencing a mail-order pharmacy mandate and roughly, half believed the mandate negatively impacted their medical care.

Respondents reporting a mail-order pharmacy mandate generally were less satisfied with pharmacy services, felt the pharmacy cared less about their health/wellbeing, collaborated less with medical providers, found the ease/urgency of refilling more challenged, and pharmacist access/knowledge were lacking based on scoring comparisons for mail-order pharmacy (Table 2B). These data portray negative patient perspectives and experiences concerning mail-order pharmacy mandates. This aspect further highlights that mail-order pharmacy mandates may themselves may be an adherence barrier, particularly with PWH, and emphasizes that patients prefer to choose which pharmacy setting works best for them individually. Insurance payers should consider the removal of mail-order pharmacy mandates for ART prescriptions to be more accommodative of all patients' circumstances and to help with progression of the initiatives laid out by the Ending the HIV Epidemic [32]. While this may lead to a short-term cost increase in prescription expenditures, the benefits of enhanced adherence and subsequent better health outcomes should lead to cost savings for payers long-term.

## Limitations

We recognize that our study has limitations. This study used survey methods for data collection, which may have included reporting bias. The generalizability of our data is limited by our small sample size, the respondents were all Midwestern, and the majority were White, commercially insured men. Further, non-English speaking patients were excluded from study participant due to IRB consenting requirements. Thus, our study did not capture the impact of language barriers on pharmacy experience. Lastly, our results may not account for other potentially confounding variables such as living in rural or urban areas, housing instability, poverty level, and lack of consistent access to telephone or electronic communication.

## Conclusions

In our small Midwestern cohort, PWH and patients using PrEP preferred local pharmacies for their ART prescription services versus mail-order pharmacies. Respondent's ranked refilling ease as most important and half believed mail-order pharmacy mandates impacted their medical care. Insurance payers should consider removing restrictive policies such as mail-order pharmacy mandates to allow patients to choose their preferred pharmacy setting for ART refills.

## Supporting information

**S1 Table. Sub-group analysis of associations with pharmacy preference.**
(PDF)

## Author Contributions

**Conceptualization:** Yadi Liu, Renae Furl, Joshua P. Havens.

**Data curation:** Yadi Liu, Joshua P. Havens.

**Formal analysis:** Yadi Liu, Elizabeth Lyden, Joshua P. Havens.

**Methodology:** Renae Furl, Joshua P. Havens.

**Project administration:** Joshua P. Havens.

**Validation:** Joshua P. Havens.

**Visualization:** Elizabeth Lyden, Joshua P. Havens.

**Writing – original draft:** Yadi Liu, Joshua P. Havens.

**Writing – review & editing:** Yadi Liu, Elizabeth Lyden, Renae Furl, Joshua P. Havens.

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
