## [Decision Letter · Decision Letter 0]

28 Apr 2023

Patient Perspectives of Antiretroviral Pharmacy Services: A Cross-Sectional Cohort Study

PONE-D-23-02690

Dear Dr. Haven

We’re pleased to inform you that your manuscript has been judged scientifically suitable for publication and will be formally accepted for publication once it meets all outstanding technical requirements.

Kind regards,

Varsha Bangalee, PhD

Academic Editor

PLOS ONE

Reviewers' comments:

Reviewer's Responses to Questions

**Comments to the Author**

1. Is the manuscript technically sound, and do the data support the conclusions?

Reviewer #1: Yes

Reviewer #2: Yes

2. Has the statistical analysis been performed appropriately and rigorously? 

Reviewer #1: Yes

Reviewer #2: Yes

3. Have the authors made all data underlying the findings in their manuscript fully available?

Reviewer #1: Yes

Reviewer #2: Yes

4. Is the manuscript presented in an intelligible fashion and written in standard English?

Reviewer #1: Yes

Reviewer #2: Yes

5. Review Comments to the Author

Reviewer #1: The abstract is well written. The introduction is well written with reference to similar studies and knowledge gap was emphasize which gave fundamental reasons for this study to be carried out. The method is replicable and the results well presented. The authors have discussed the study well enough and made sound recommendations.

Reviewer #2: A good work, Can be published without any changes.

The work is good and very much appropriate.

I recommend it for the publication without any modifications.

It will provide a students and researchers some base.

6. PLOS authors have the option to publish the peer review history of their article (what does this mean?). If published, this will include your full peer review and any attached files.

Reviewer #1: No

Reviewer #2: No

---

## [Editor Report · Acceptance letter]

11 May 2023

PONE-D-23-02690 

Patient Perspectives of Antiretroviral Pharmacy Services: A Cross-Sectional Cohort Study 

Dear Dr. Havens:

I'm pleased to inform you that your manuscript has been deemed suitable for publication in PLOS ONE. Congratulations! Your manuscript is now with our production department. 

Kind regards, 

on behalf of

Professor Varsha Bangalee 

Academic Editor

PLOS ONE